# Effects of Palm Oil Deodorizer Distillate on the Ruminal Environment of Sheep

**DOI:** 10.3390/ani14091269

**Published:** 2024-04-24

**Authors:** Diego Assis das Graças, Eziquiel de Morais, Alyne C. S. Lima, Shirley M. de Souza, Luciano F. Sousa, Diego C. Franco, Artur L. C. Silva, André G. Maciel e Silva

**Affiliations:** 1Laboratório de Engenharia Biológica, Guamá Science and Technology Park, Belém 66075-110, PA, Brazil; arturluizdasilva@gmail.com; 2Instituto Federal de Ciência e Tecnologia do Pará, Castanhal 29056-264, PA, Brazil; eziquielmorais@yahoo.com.br; 3Instituto Federal de Ciência e Tecnologia do Amapá, Campus Porto Grande, Macapá 68997-000, AP, Brazil; alynecslima@gmail.com; 4Instituto de Medicina Veterinária, Universidade Federal do Pará, Castanhal 68740-970, PA, Brazil; motta.shirley@hotmail.com (S.M.d.S.); andregms@gmail.com (A.G.M.e.S.); 5Escola de Medicina Veterinária e Zootecnia, Universidade Federal do Tocantins, Campus de Araguaína, Araguaína 77826-612, TO, Brazil; luciano.sousa@mail.uft.edu.br; 6Laboratório de Ecologia Microbiana, Instituto Oceanográfico, Universidade de São Paulo, São Paulo 05508-120, SP, Brazil; diecasfranco@gmail.com

**Keywords:** ruminal degradability, microbial diversity, POD, 16S

## Abstract

**Simple Summary:**

Numerous palm oil byproducts are already employed as livestock energetic feed supplements, and the inclusion of palm oil deodorizer distillate (POD) presents a potential addition to cost-effective feed alternatives. However, the presence of high-fatty-acid diets could alter the ruminal environment, potentially influencing the digestive efficiency. This study aimed to assess the impact of POD on the digestive system of sheep. Twenty sheep were fed different diets containing varying amounts of palm oil deodorizer, and their rumen fluid was analyzed for the microbial composition and digestive efficiency. The microbial community in the rumen was primarily composed of Bacteroidetes and Firmicutes, with minor changes occurring when the POD was added to the diet. Up to 25 g of POD per kilogram of dry matter negatively affected the ruminal fermentation or apparent digestibility, but not to a high degree, indicating that POD is a viable and cost-effective supplement for sheep diets. However, higher POD levels hindered the breakdown of fibrous material in the rumen and overall dry matter digestibility. These findings suggest that while POD can be a beneficial and economical option for sheep nutrition, carefully considering its dosage is crucial for maintaining optimal digestive function and nutrient utilization, offering valuable insights for sustainable and efficient livestock feeding practices.

**Abstract:**

This study aimed to assess the impact of palm oil deodorizer distillate (POD) on the ruminal environment, including (i) microbial community, (ii) ruminal degradability, and (iii) apparent digestibility in sheep. The data used were derived from twenty rumen-cannulated sheep fed five isoproteic and isofiber diets based on elephant grass (*Pennisetum purpureum* Schum. cv. Roxo) silage supplemented with 0, 25, 50, 75, or 100 g kg^−1^ POD on a dry matter (DM) basis. Rumen fluid samples were collected three hours after feeding directly from the ventral sac of the rumen via a cannula and then subjected to DNA extraction, which was subsequently used for 16S rDNA amplification, followed by sequencing and diversity analysis. In this study, the microbial diversity was dominated by Bacteroidetes and Firmicutes, followed by Euryarchaetoa, Actinobacteria, and Tenericutes, in the ruminal environment, and was slightly modified when supplemented with the POD up to 100 g/kg (10%), leading to only a slight decrease in the diversity index. The ruminal degradability, ruminal fermentation parameters, and apparent digestibility were slightly compromised by the inclusion of up to 25 g of POD per kg of DM, and larger inclusions interfered with the ruminal degradability of fibrous fractions and the apparent digestibility of dry matter. This lipid supplement showed good results for feeding sheep and is an inexpensive and abundant alternative in the regional market.

## 1. Introduction

Palm oil (*Elaeis guineensis*, Jacq) has emerged as the most extensively cultivated vegetable oil globally and is extensively used in industry, particularly in the food industry. According to Abrapalma [1], 72 million tons of palm oil are produced worldwide, and in Brazil, Pará State contributes approximately 85% of the national production of this oil.

In edible oil production, refining encompasses the deodorization process, which produces the byproduct known as palm oil deodorizer distillate (POD) [2]. This byproduct has a low market price but high availability, as well as characteristics that possibly make it environmentally suitable for use in animal feed.

POD has a persistent consistency at room temperature and, like palm oil, has high concentrations of saturated fatty acids, such as 42% palmitic acid and 5% stearic acid, and unsaturated fatty acids, such as 43% oleic acid and 10% linoleic acid [3]. However, many studies were conducted to evaluate the potential utility of palm oil or other byproducts, such as palm oil or palm kernel cake, in ruminant nutrition, whereas only a few studies evaluated the potential of POD for this purpose [4].

The microbial community plays a major role in the digestion process of ruminant animals, as it is responsible for organic matter conversion into short-chain carbonic matter [5]. The relationship between the animal diet and the rumen microbiome was the aim of several studies [5], including studies on the effects of the type of cellulose [5], the inclusion of probiotics [6], and lipid intake [7]; however, to the best of our knowledge, no studies have investigated palm oil distillates. We aimed to evaluate the effects of POD on the microbial community structure, ruminal degradability, and apparent digestibility in sheep.

## 2. Materials and Methods

### 2.1. Animals, Experimental Design, and Diets

This study was performed in Castanhal, Pará, Brazil (1°17′49″ S/47°55′19″ W). Twenty crossbred sheep, each fitted with a rumen cannula, weighed 35.8 ± 9.46 kg. They were housed individually in stalls measuring 0.8 m × 1.5 m with cement floors, lined with sawdust, and equipped with feeding and drinking troughs. The experiment comprised five treatments, each involving four animals. These animals were fed five diets, all of which were isonitrogenous, isofibrous, and formulated with elephant grass (*Pennisetum purpureum* Schum. cv. Roxo) silage. The diets were supplemented with 0, 25, 50, 75, or 100 g kg^−1^ of POD (based on the total dry matter (DM)), denoted as POD0, POD25, POD50, POD75, and POD100, respectively (Appendix A).

To produce the silage, the grass was harvested after 90 days of regrowth. The concentrate consisted of a soybean meal, a mineral mix, and ground corn grain. The daily diet was provided to the sheep at a ratio of 1.5 g per kg of body weight, with intake restricted according to National Research Council (2007) [8] guidelines, maintaining a forage-to-concentrate ratio of 1:1 based on DM. POD was blended with the concentrates to ensure even distribution and intake. Feed was administered in two equal portions. Before data collection, a 21-day adaptation period was provided for the animals to acclimate to the diets and experimental conditions. The fatty acid profile of POD is shown in Appendix A following procedures described by Dirksen (1990) [9].

### 2.2. Ruminal Degradability

The silage was dried in a forced ventilation oven at 55 °C for 72 h and ground with a Willey mill with 5 mm mesh sieves. Duplicate samples (4 g DM) were weighed, placed in 12 × 5 cm nylon bags with a porosity of 50 μm [10], and introduced into the rumen. Following incubation periods of 6, 12, 24, 48, 72, and 96 h, the nylon bags were retrieved from the rumen, immediately immersed in cold water, washed under running water until colorless, and then transferred to a forced ventilation oven at 55 °C until reaching a constant weight. Disappearance at time zero was determined by washing the unincubated bags using the same procedure as described for the incubated bags.

Dry matter (DM), organic matter (OM), crude protein (CP), neutral detergent fiber (NDF), and acid detergent fiber (ADF) degradability parameters were fitted according to the model described by Orskov et al. (1979) [10]. To estimate the effective degradability (ED), we used the formula according to Morais et al. (2023) [11], and the statistical analysis was performed via a randomized design [12] using a nonlinear mixed-effects model with PROC NLMIXED in SAS^®^ (Statistical Analysis System, version 9.2.) [13,14].

### 2.3. Apparent Digestibility

Indigestible neutral detergent fiber (iNDF) was determined using a non-woven textile of in situ ruminal incubation with the supplied feedstuffs, residual feed, and feces [15,16]. We conducted sample incubation in triplicate in the rumen of a cannulated buffalo according to previous studies [12,17]. The apparent digestibility assay was developed as a completely randomized design. The data were analyzed with the general procedure of linear models according to Morais et al. (2023) [12].

### 2.4. Ruminal Fluid Sampling and Fermentation Pattern

The ruminal content was sampled using a rumen cannula at 0, 1, 3, 6, 9, and 11 h after feeding. The pH, NH_3_-N, and methylene blue reduction time (MBR) were determined according to previous studies [12,18].

The population of protozoa was determined in ruminal fluid collected via a ruminal cannula 3 h after feeding. One milliliter of ruminal fluid was stored in 9 mL of 10% formaldehyde solution at a dilution of 1/10. In contrast, three drops of the bright green solution were added to each sample, and counting was performed in a Neubauer chamber with a 0.1 mm depth under a binocular optical microscope using a 10 times objective lens [19].

### 2.5. Ruminal Fluid Sampling, DNA Extraction, and PCR

To assess the genetic diversity of the ruminal community, 60 mL of rumen fluid samples were collected three hours after feeding directly from the ventral sac of the rumen via the cannula, and then a 3 mL subsample was placed in a tube containing RNA holder (BioAgency Biotecnologia Ltd.a., São Paulo, SP, Brazil) (1:1) to preserve the material and frozen for further analysis.

Rumen fluid samples obtained from animals subjected to the same diet were pooled, severely vortexed for homogenization, and then centrifuged for 15 min at 4000× *g*, after which the supernatant was discarded. The pellets were subjected to DNA extraction using an UltraClean Microbial DNA Isolation Kit (MoBio, Carlsbad, CA, USA) according to the manufacturer’s instructions. Therefore, DNA was used as a template for 16S rDNA amplification using the universal primers 515F and 806R, which were tagged with barcodes for each diet. PCR was carried out in a final volume of 50 µL containing 1× buffer, 0.2 mmol/L dNTPs, 4.0 mmol/L MgCl_2_, 0.5 µmol/L each primer, 0.7 U Taq DNA polymerase, and 0.3 mg/mL bovine serum albumin. The thermal cycling conditions consisted of an initial step of 94 °C for 2 min, followed by two cycles of 94 °C for 1 min, 48 °C for 1 min, and 72 °C for 1 min; two cycles of 94 °C for 1 min, 52 °C for 1 min, and 72 °C for 1 min; and 26 cycles of 94 °C for 1 min, 56 °C for 1 min, and 72 °C for 1 min.

### 2.6. DNA Sequencing and Phylogenetic Analysis

Equimolar amounts of the amplicons from each sample labeled with barcodes were placed into a mixture and subjected to sequencing. The libraries were constructed and sequenced on PGM™ Ion using a PGM™ Ion Sequencing 400 Kit and finally deposited on an Ion 318^TM^ Kit v2 chip according to the manufacturer’s protocol (Life Technologies, Carlsbad, CA, USA).

Low-quality reads (maximum error probability of 0.5), smaller than 200 bp, and chimera were removed using USEARCH tools [20]. Sequences were clustered at 97% similarity and chimera were filtered, and singleton reads were removed. Taxonomy was assigned to each operational taxonomic unit (OTU) by performing BLAST searches against the SILVA database v.132 [21] with a maximum E-value of 1 × 10^−5^.

The phyloseq R package (https://www.r-project.org/, accessed on 12 October 2020) was used to calculate the alpha diversity indices and differences in alpha diversity estimates between groups of samples were tested using Student’s *t*-test. The OTU table was normalized for beta diversity analysis using cumulative sum scaling (CSS) [22]. Beta diversity was calculated using weighted UniFrac.

### 2.7. Residual and Sample Chemical Analyses

Chemical analyses of the incubation residues, feed, and feces samples involved pre-drying in a forced air ventilation oven (at 55 °C for 72–96 h), followed by milling using a knife mill equipped with 1 mm mesh sieves. Samples were analyzed for ash, crude protein (CP), ether extract (EE), neutral detergent fiber (NDF), and acid detergent fiber (ADF) [17] based on DM content using heat-stable amylase and without sodium sulfite.

## 3. Results

### 3.1. Ruminal Degradability

The effective degradability coefficients of the DM, OM, and NDF decreased with the inclusion of the POD in the diet (Appendix A). The degradability results for the CP demonstrated alterations in both the soluble and potentially degradable fractions. There was a gradual decrease in the soluble fraction and a gradual increase in the potentially degradable fraction (Appendix A), which led to changes in the pattern of the degradation curve, as well as in the degradation itself. However, the ED was not modified by the POD inclusion (Appendix A).

The POD used in the present study contained 51.76% saturated fatty acids, of which 46.05% were palmitic fatty acid (C16:0), 0.25% lauric acid (C12:0), 0.72% myristic acid (C14:0), and 4.74% stearic fatty acid (C18:0); the unsaturated FAs comprised 39.06% of oleic acid (C18:1), 8.37% linoleic acid (C18:2), and 0.22% linolenic acid (C18:3).

The use of POD linearly decreased the digestibility of the DM, OM, and fiber fractions (NDF and ADF). The digestibility of the EE increased linearly with the inclusion of the POD (Table 1), the digestibility of the CP was not influenced by the inclusion of the POD.

The Williams test, which indicated the point of regression where the values became significantly different from the control group (POD0), showed that the DM started to be affected by the inclusion of 24.9 g kg^−1^ of POD, whereas for the OM, the effect was observed from the inclusion of 53.4 g kg^−1^ of POD (Table 1). The digestibility of the fibrous fractions NDF and ADF did not differ from that of the control diet with the inclusion of 44 and 38.5 g kg^−1^ of POD, respectively. The digestibility of the EE started to be different from the control POD from 24.4 g kg^−1^ of the inclusion of the lipid source.

### 3.2. Microbial Diversity

According to the evaluation of the population of microorganisms, sequencing revealed approximately 4 million reads, with an average size of 304 bp. The observed diversity (richness) of the samples (Table 2) was relatively similar; the diet without supplementation had the richest diversity, and the diet with the highest lipid content (POD100) had the lowest diversity. This was also observed through the interpretation of the Shannon index, which confirmed a trend of a discrete decrease in diversity with diets with higher lipid concentrations.

The most abundant phyla were Bacteroidetes and Firmicutes, followed by Euryarchaeota, Actinobacteria, and Tenericutes (Appendix A). The samples with the highest lipid concentrations (POD75 and POD100) were more similar, as revealed by the dissimilarity analysis (Figure 1), and differed from the other samples and the sample without supplementation. Supplementation had a small effect on the microbial community since the distances were based on a dissimilarity scale of 0.05.

## 4. Discussion

The dry matter (DM) represents the portion of food that encompasses all other fractions, with the fibrous fraction being the main component in forages. Changes in this fraction might influence the degradability of the DM and OM. Including the POD in the diet affected the degradation kinetics of the DM and NDF. This was possibly due to the negative effects of the lipids on the degradation of the fibrous fraction since this was the fraction most affected by the inclusion of lipids in the diet [23].

Given its high content of saturated fatty acids (more than 50%) (Appendix A), which is associated with less negative effects on fiber digestion [24], the hypothesis was that POD could be included at levels higher than 70 g kg^−1^ without causing changes in the DM digestibility of nutrients, especially on fiber (NDF and ADF); however, this was not supported.

The apparent digestibility of the EE had a linear positive effect of 1.16% for each additional 1% of the POD, with an estimated digestibility of 84% (at 10 g kg^−1^ DM inclusion), which is considered normal for lipids (70 to 90%) [25]. However, for the DM and OM, a negative linear effect was observed, with reductions of 0.66 and 0.67%, respectively, for each inclusion percentage unit. These results were probably due to the reduced digestibility of the fibrous fractions [26], as discussed below, which comprised most of the DM and OM.

For the NDF and ADF fibrous fractions, there was a more expressive negative linear effect than for the DM and OM fractions; the reductions were 0.95 and 0.91% for each POD inclusion unit, respectively. The results of the present study do not corroborate those observed by Raiol et al. (2012) [3], who did not find a difference in the apparent digestibility of NDF in lambs subjected to diets containing 50 g of POD/kg of DM.

The ruminal fermentation parameters N-NH_3_, pH, and MBR (Appendix A) indicated normal microbial activity for all treatments. The concentrations of N-NH_3_ did not differ when including the POD in the diets. The mean N-NH_3_ level averaged 14.62 mg/dL, reaching its peak at 20.23 mg/dL one hour post-feeding and dropping to a minimum of 9.51 mg/dL twelve hours later. Even at 12 h after feeding, the concentrations were greater than 5 mg/dL, which is the minimum for proper ruminal fermentation [27,28].

Including lipids in the diet can decrease the population of protozoa and other microorganisms, including methanogenic and Gram-positive bacteria, with the latter being the group containing fiber degraders [28]. The decrease in protozoa population could be detected when the concentration increased from 25 to 50 g kg^−1^, increasing the EE from 59.6 to 83.4 g kg^−1^, respectively.

In general, the use of oil or fat in ruminant diets is associated with reduced populations of protozoa in the rumen [29]. The protozoa function in the rumen as fermenters of starch and lactate, which helps to regulate the pH. They have predation activity on bacteria [30], which results in nitrogen recycling [31]. They also participate, in a symbiotic manner, in the biohydrogenation of unsaturated fatty acids. However, their real importance in the ruminal environment is controversial, and they might even be dispensable [32]. Improvements in animal performance, such as increased protein flow for the abomasum and improved feed conversion, are reported when protozoa are eliminated [33]. In addition, the elimination of or reduction in the protozoan population in the rumen may also be associated with a reduction in the methanogenic population since these microorganisms have an intimate relationship with ruminal protozoa [29].

The methylene blue reduction time (MBR) assay facilitates real-time assessment of microbial activity by measuring its redox potential. The time considered normal for a complete reduction in ruminal fluid varies from 3 to 6 min [18].

Some studies investigated the effects of different lipid sources on ruminal metabolism and demonstrated how cellulolytic species and strains are affected by unsaturated fatty acids [24,33]. Recently, molecular approaches have revolutionized our understanding of the role of fats, the complex environment of the rumen, and the interactions between the two. In this context, a study evaluated the effect of different sources of lipids, such as whole soybean, whole kapok seed, and cracked oil palm fruit, on the microbial populations in the rumen of cattle. The effects of dietary oil on the abundance of total bacteria and *F. succinogenes* are high in cattle fed palm oil fruit, which might be related to the fact that the fats in the diet might exert low inhibitory and/or coating effects on microorganisms [34].

The core microbiome is composed of 15 phyla, and more than 80% of the sequences can be attributed to Bacteroidetes and Firmicutes, like what was observed in a study with beef cows, where it was possible to observe these phyla as the most abundant [35]. At the genus level, the most abundant sequences were assigned to the *Eubacterium coprostanoligenes* group, *Christensenellaceae R−7 group*, *Rikenellaceae RC9 gut group, Staphylococcus, Enterococcus*, and *Ruminococcus.* These microbes are abundant in the sheep rumen [36], which is concordant with our study; involved in fiber digestion; responsible for proteolysis, carbohydrate degradation, and amino acid fermentation to acetate [37].

Notably, the composition of the microbial community at the phylum and genus levels slightly changed, showing that the ruminal environment remained stable, even at the highest level of lipid supplementation. In addition, there was a decrease in Tenericutes and Verrucomicrobia, which was compensated for by the growth of Actinobacteria, as the lipid percentage of the diet increased. In a study with steers supplemented with soybean oil [38], this diet greatly affected bacterial diversity, whereas the methanogenic diversity did not change much. Kairenius and colleagues [39] did not observe large changes in the bacterial populations of cows supplemented with different types of isolated or combined oils.

The most abundant orders were Bacteroidales and Clostridiales, which are related to the process of biohydrogenation of unsaturated compounds in the diet [40]. However, to increase this concentration in ruminant products, intensive biohydrogenation carried out by ruminal microorganisms must be avoided, allowing for the availability of unsaturated fatty acids in the small intestine. The high prevalence of the orders Bacteroidales (>40%) and Clostridiales (>35%) is frequent in studies using highly concentrated diets for lambs [41].

The order Methanobacteriales was the only representative of the methanogens, remaining below 10% in abundance, with a slight decrease with the highest oil inclusion levels. The addition of soybean oil alters the abundance but not the diversity of methanogens [40]. When dairy goats are supplemented with hemp oil or flaxseed, no significant differences are found between the genera of archaea between treatments [42].

## 5. Conclusions

The microbial diversity in the ruminal environment was slightly modified when supplemented with POD up to 10%, leading to only a slight decrease in the diversity index. The inclusion of up to 25 g of POD per kg of DM did not compromise ruminal degradability, fermentation parameters, or apparent digestibility. However, higher amounts interfered with the degradability of fibrous fractions and the digestibility of dry matter. This underscores the typical advice to limit dietary fat to a maximum of 6% to enhance fiber digestion. This lipid supplement, which is rich in saturated fatty acids, showed good results for ruminants, but the influence of gene activity should be investigated to better understand the influence of the lipid diet on the activity of the microbial community.

## Figures and Tables

**Figure 1 animals-14-01269-f001:**
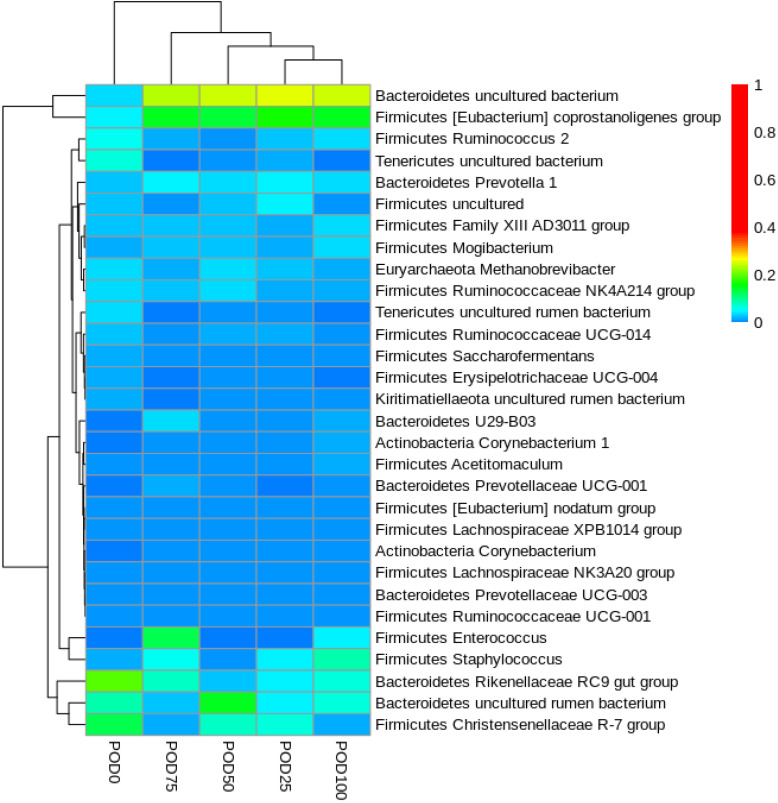
Heatmap showing dissimilarities (beta diversity) between samples and the most abundant genera.

**Table 1 animals-14-01269-t001:** Apparent digestibility coefficients in sheep fed with POD.

Variable	Apparent Nutrient Digestibility (g kg^−1^ DM)									
	POD0	POD25	POD50	POD75	POD100	RE	P	R^2^	SEM	D
DM	61.92	59.40	59.82	56.02	55.40	Y=61.79−0.66X	0.0001	89.90	0.71	24.9
OM^2^	63.49	61.31	61.07	57.67	56.96	Y=63.44−0.67X	0.0001	94.06	0.70	53.4
CP	67.75	65.24	67.82	65.94	65.77	NS	0.6060	-	0.85	>100
EE^1^	71.84	78.40	82.37	83.21	83.95	Y=74.15+1.16X	0.0009	83.71	1.34	24.4
NDFap	55.12	52.87	49.42	46.53	44.88	Y=54.80−0.95X	0.0000	97.42	1.12	44.0
ADFap	53.33	50.41	46.94	44.08	43.68	Y=52.61−0.91X	0.0000	99.76	1.10	38.5

Abbreviations: DM—dry matter; OM^2^—organic matter; CP—crude protein; EE^1^—ether extract; NDFap—neutral detergent fiber corrected for ash and protein; ADFap—acid detergent fiber corrected for ash and protein; P—*p*-value; RE—regression equation; R^2^—determination coefficient; NS—not significant; SEM—standard error of the mean; D—Williams test result expressed in g kg^−1^ of POD.

**Table 2 animals-14-01269-t002:** Sequencing and alpha diversity results for all treatments at 97% similarity.

Level	Raw Reads	OTUs	Chao1	Shannon
POD0	318,760	1049	1191	9.73
POD25	276,510	988	1159	9.65
POD50	233,493	929	1072	9.54
POD75	195,022	844	947	9.41
POD100	218,803	840	940	9.4

## Data Availability

DNA sequences were submitted to the SRA under Bioproject ID PRJNA487521.

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
