# Peer review of "Effects of Palm Oil Deodorizer Distillate on the Ruminal Environment of Sheep"

_animals, 2024, doi:10.3390/ani14091269_

Round 1
Reviewer 1 Report
Comments and Suggestions for Authors
I have reviewed the article entitled “Effects of a Diet Rich in Palm Oil Deodorizer Distillate on the Ruminal Environment of Sheep”
I consider it to be a good article despite its limited contribution to knowledge. While the aim of the study is to evaluate the effects of POD inclusion on the ruminal environment, microbial community structure, ruminal degradability, and apparent digestibility to promote its use, the importance of the study is that it addresses relevant issues such as finding alternative feeds to reduce production costs in sheep farms, and the efficient use of resources, with a particular emphasis on incorporating local inputs within a sustainable framework.
To achieve the study's aim, the authors analyzed the impact of Palm Oil Deodorizer Distillate (POD) on the digestive system of sheep, utilizing 20 fistulated sheep according to a Completely Randomized Design with 5 treatments and 4 repetitions per treatment. Five treatments were tested consisting of 0, 25, 50, 75, and 100 g/kg of POD on dry matter basis supplemented in elephant grass (Pennisetum purpureum) silage. Ruminal fluid extraction was performed, DNA was obtained, and microbial diversity was studied. The results showed that when POD was used at high doses, it interfered with ruminal degradability and dry matter digestibility.
The main contribution of the study is that the use of POD at low levels (les than 25%), is a good alternative for sheep feeding, since it represents an inexpensive and highly available feed in regional markets.
Author Response
Thank you for your contribution.
Reviewer 2 Report
Comments and Suggestions for Authors
Please refer to the attachment

Minor editing required
Author Response
Dear reviewer,
Reviewer 2
Here is a point-by-point response for all suggestions:
- General comments
- Title change: Done.
- The link between M&Ms and Results: Improved.
- Specific comments
- Line 13: As an introduction of the summary, the authors could indicate that given then several palm oil byproducts already used as livestock feed supplements, POD could be added to the repertoire of low-cost feed alternatives. However, diets rich in fatty acids could modify the ruminal environment and affect digestive efficiency. They can rephrase to be succinct.
R: We modified the text.
- Line 26: Isn’t the microbial community part of the rumen environment?
R: We modified the text.
- Line 29: Please confirm that the sampling times agree with what is mentioned under the materials and methods section.
R: Yes, the sampling time agrees with what is mentioned under the material and methods. We used 3 hous after feeding to sample ruminal fluid to protozoa population and genetic diversity and 0, 1, 3, 6, 9 an 11 h for pH, NH3-N, MBR evaluations.
- Line 83: Please explain at what point baseline data was collected. For example, what were the base ruminal parameters considered for this study?
R: We used the control group (a group that didn´t received POD as the base for ruminal and digestive parameters.
- Line 96: While DM was defined at its first use in the text (line 29), OM, CP, NDF, and ADF were not.
R: We modified the text.
- Line 108-110: Please check the grammar and rephrase for clarity.
R: We modified the text.
- Line 163-164: Please add the reference for R.
R: We added.
- Line Lines 267-267 contradict lines 63-64.
R: We modified the text..
- Line 26: Isn’t the microbial community part of the rumen environment?
R: We modified the text.
- Line 29: Please confirm that the sampling times agree with what is mentioned under the materials and methods section.
R: We corrected.
- Line 96: While DM was defined at its first use in the text (line 29), OM, CP, NDF, and ADF were not.
R: Modified.
- Line 108-110: Please check the grammar and rephrase for clarity.
R: Done.
- Line 163-164: Please add the reference for R.
R: We added.
- Line Lines 267-267 contradict lines 63-64.
R: We modified the text.
- Line 111: Please confirm that ethical approval also included use of the Buffalo in this study.
R: Confirmed. Buffaloes were included in the ethical approval.
- Line 112: Please confirm that “palm cake” is not “Palm kernel cake”.
R: Corrected. It's palm kernel cake.
- Line 140: Why were the samples pooled? Could this lead to confounding? What would be the likely effects on statistics obtained?
R: We used this sudo replication strategy due to financial restrictions. We agree this is not the best choice, and it's a limitation of our study, but the sequencing results were deep enough to achieve diversity saturation.
- Line 163: How were the indices calculated?
R: We used the phyloseq package on R. We modified the text.
- Line 175-176: At what point did the design change. I can’t seem to find this in the design description under the subheading “2.1. Animals, experimental design, and diets”.
R: Modified. Please, see the lines 179-181 from the corrected version. As shown on this topic, the first paragraph describes the design of the in situ degradability and the second paragraph describes the digestibility design and analysis.
- Line 174: Unless implied, please provide the rationale for the choice of models used here. This section is difficult to follow and should be re-written.
R: Rewritten.
- Line 223: This should have been included under materials and methods.
R: Included.
- Line 257-259: Re-write for clarity. Is the POD effect on the digestibility of EE or it is the digestibility of EE that is the effect on POD?
R: EE is the fraction soluble in Ether, this analysis indicates the fatty acid content of a feed. POD, as a fatty acid source, influenced the diet EE. The POD, as a dietetic EE source, influences DM digestibility, especially fibrous fraction.
- Line 260-262: requires firming up with relevant supporting literature.
R: Reference added.
- Line 305-314: requires firming up with relevant supporting literature. While the author refers to several studies, only one reference is provided for this paragraph.
R: Text modified and references added.
- Line 317: what does “forage-fed cutting cows” mean?
R: Beef cows. Text modified.
- Line 328-331: Is soy oil the best comparison for POD? Do their nutrient profiles support such a comparison?
R: Despite the different nutrient profiles, some studies make such comparisons. But we can remove if you recommend it.
- Line 333-335: reference?
R: Text removed.
- Line 340-343: This has already been mentioned elsewhere in the discussion.
R: Text removed.
- Line 345-349: It is not clear how this section fits into the study. Were the authors comparing POD and soy oil?
R: Same answer from question in Line 328-331.
- Line 350: Please draw a conclusion based on the content of this study. Also provide insights useful for future prospects with POD.
R: Rewritten.
Minor text changes were all done and excluded from the list above.
Thank you again for your contributions.
Reviewer 3 Report
Comments and Suggestions for Authors
This manuscript describes an experiment evaluating increasing inclusion levels of palm oil distillate in the diet of sheep. The primary concern that I have is with the description of the results. The language is vague and results are not stated clearly as to be understood by the reader. There are some discrepancies between the statistical results and interpretation of results. For example, it is stated that digestibility was not impacted up to 25 g/kg however, the statistical analysis indicates that there is a linearly relationship between inclusion level and digestibility meaning that digestibility was impacted over the entire range of inclusion levels. The Results section needs to be rewritten and the interpretation of results needs to match statistical analysis of results.
Specific comments are in the attached document.

English language is good except that descriptions of results are often vague.
Author Response
Dear reviewer,
Thank you for your contributions.
Here is a point-by-point response for all suggestions:
L36: I am not sure this is correct interpretation of the data becuase there was a linear effect of POD inclusion level.
R: You're right. We corrected.
L55: these are not saturated fatty acids
R: We modified the text
L109: is this performed on the TMR or individual feedstuffs?
R: It was performed on the individual feedstuffs.
L136: This is a very small sample of fluid to collect. I would expect to collect larger amount then subsample for the laboratory analysis.
R: A 60 mL sample was collected, and then a subsample of 3 mL was placed in RNA holder. Text modified.
L198: Overall the description of the results is difficult to understand. There needs to be more explanation for the reader to follow.
R: This topic was massively reworked.
L201: in Table S3, I do not see the effective degradability increasing at POD100.
R: Agreed. We modified the text.
L203-206: I think this paragraph makes sense for the CP results, but the DM and OM results do not follow this pattern and they have a different pattern of results than CP with increasing POD
R: Agreed. Corrected.
L212: is this alpha or gamma linolenic acid?
R: Unfortunately, we were unable to acquire the desired answer. Despite subcontracting the analysis, the results yielded only "linolenic acid," leaving ambiguity that it might encompass both possibilities.
L213-215: This needs more explanation. It needs to help the reader interpret what 2.49% and 5.34% means.
R: Rewritten.
L254-256:The discussion of dietary fat content needs to be expanded. This is the primary basis for differences in fermentation parameters especially the linear decrease in digestibility.
R: Reworked.
L267: is this supposed to be 'NDF in lambs'? digestibility of lambs does not make sense
R: We modified the text.
L276-279:This does not make sense to me. What are intense side effects? What was detected in 76.6% of the population? Population of what?
R: Rewritten.
L294-297: this paragraph should move up with the previous paragraph to make a single paragraph
R: Agreed. We modified this paragraph.
L300-301: what is perfect microbial activity? what disagreement?
R: We modified the text.
L354: need to mention that this fits the general recommendation for total fat content in ruminant diets to be 6% or less for maximum forage digestion
R: Agreed. We included this information in.
Minor text changes were all done and excluded from the list above.